# Switchable stimulated Raman scattering microscopy with photochromic vibrational probes

Jianpeng Ao[1,4], Xiaofeng Fang[2,4], Xianchong Miao[1], Jiwei Ling[1], Hyunchul Kang[3], Sungnam Park [3], Changfeng Wu [2✉] & Minbiao Ji [1✉]

Photochromic probes with reversible fluorescence have revolutionized the fields of single molecule spectroscopy and super-resolution microscopy, but lack sufficient chemical specificity. In contrast, Raman probes with stimulated Raman scattering (SRS) microscopy provides superb chemical resolution for super-multiplexed imaging, but are relatively inert. Here we report vibrational photochromism by engineering alkyne tagged diarylethene to realize photo-switchable SRS imaging. The narrow Raman peak of the alkyne group shifts reversibly upon photoisomerization of the conjugated diarylethene when irradiated by ultraviolet (UV) or visible light, yielding "on" or "off" SRS images taken at the photoactive Raman frequency. We demonstrated photo-rewritable patterning and encryption on thin films, painting/erasing of cells with labelled alkyne-diarylethene, as well as pulse-chase experiments of mitochondria diffusion in living cells. The design principle provides potentials for super-resolution microscopy, optical memories and switches with vibrational specificity.

---

[1] State Key Laboratory of Surface Physics and Department of Physics, Human Phenome Institute, Multiscale Research Institute of Complex Systems, Academy for Engineering and Technology, Key Laboratory of Micro and Nano Photonic Structures (Ministry of Education), Fudan University, Shanghai, China. [2] Department of Biomedical Engineering, Southern University of Science and Technology, Shenzhen, China. [3] Department of Chemistry and Research Institute for Natural Science, Korea University, Seoul, Korea. [4] These authors contributed equally: Jianpeng Ao, Xiaofeng Fang. ✉email: wucf@sustech.edu.cn; minbiaoj@fudan.edu.cn

Photochromism induced by reversible photoisomerization yielding different spectral properties have broad applications in optical memories, switches and actuators[1,2]. In particular, photoswitchable fluorescent molecules have revolutionized modern light microscopy with spatial resolution far beyond diffraction limit, such as the reversible switchable/saturable optical fluorescent transition (RESOLFT) and single-molecule localization nanoscopy[3–6]. While the electronic transitions of fluorescent molecules have the advantage of ultra-brightness up to single molecule sensitivity, the intrinsic broad/overlapping spectral feature has limited the resolving power of probe species.

Vibrational transitions with much narrower spectral linewidth have superior chemical specificity to resolve different molecular species, structural changes and dynamics[7]. With the development of stimulated Raman scattering (SRS) microscopy, conventionally weak Raman scattering could be amplified by coherent and nonlinear optical process ($\sim10^3-10^5$ gain), enabling rapid label-free chemical imaging for living cells and tissues[7–13]. Further coupled with engineered Raman probes, SRS has demonstrated various imaging capabilities competing with fluorescence based microscopies[14,15]. These include super-multiplexed imaging using polyynes and alkyne-tagged dyes to resolve more than 20 labels simultaneously[16,17], and single molecule vibrational sensitivity with stimulated Raman excited fluorescence[18,19]. However, the much smaller cross-sections and the photostable nature of local bond vibrations have hindered Raman probes to obtain the intriguing properties of fluorescence molecules, including stochasticity[5], photosaturation[20] and photoswitchability[3], which are the key components for most fascinating imaging techniques.

We report a design of photo-switchable vibrational probe by introducing the strong bioorthogonal Raman tag - alkyne group into the photochromic diarylethene (DTE). When photo-isomerization occurs under UV irradiation, the ring-closing reaction changes the electronic structure and redshifts the absorption spectra of the DTE unit, which not only induces a relatively large Raman frequency shift of the conjugated alkyne group, but also enhance its SRS intensity through electronic pre-resonance effect. Consequently, a resolvable photoactive Raman peak could be reversibly turned on/off with controlled UV/visible light. In the current work, we characterized the basic vibrational photo-switching properties of the engineered Raman probes, and applied them to rewritable information patterning, as well as controlled activation, erasing and chasing in living cells with SRS microscopy.

## Results

### Synthesis and spectral characterization of photo-switchable Raman probes.
Alkyne derivatives tagged diarylethenes were synthesized according to the literature as illustrated in Supplementary Fig. 1a and Supplementary note 1[21–23]. DTE is a well-known photochromic dye with outstanding thermal stability of both isomers, which could be converted reversibly by UV/visible irradiation, generating closed/open-ring structures with different absorption spectra. These molecules present rapid response time, high photoisomerization quantum yield and good fatigue resistance[1,24]. The overall electronic properties of the synthesized DTE-TMS and DTE-Ph molecules resemble that of the native DTE with slight changes of conjugation strength as shown in the UV-Vis spectra of both isomers (Supplementary Fig. 1b). Density functional theory (DFT) calculation gives the optimized geometries of open/closed isomers (Supplementary Fig. 2a). In the closed-ring isomer, the two thiophene rings in both molecules form a planar conjugated structure, whereas the open-ring isomer forms a twisted 3D geometry.

We first characterized the spontaneous Raman response of the photoisomerization effect of the DTE-alkyne derivatives. With visible (633 nm) and 785 nm pump, a single and sharp Raman peak around 2214 cm$^{-1}$ could be detected in DTE-Ph sample (Supplementary Fig. 3), which is attributed to the symmetric stretching mode of the two alkyne groups in the open-ring form. However, the UV induced closed-ring isomer could be hardly detected, because the continuous irradiation of the Raman pump laser locks the molecules in the open-ring structure. Only under longer wavelength (1064 nm) excitation can we stably measure the Raman spectra of the closed-ring isomers after UV irradiation. As we can see, a red-shift of the triple-bond Raman frequency occurs with UV induced photo-cyclization (Fig. 1a), which is also confirmed with DFT calculation (Supplementary Fig. 2b). Both the absolute vibrational frequency ($\Omega$) and the amount of frequency shift ($\Delta\Omega$) are sensitive to the attached alkyne derivatives. For DTE-Ph, a large frequency shift ($\sim20$ cm$^{-1}$) was generated, resulting in a well-resolved new Raman band, whereas DTE-TMS generates a smaller shifted ($\sim10$ cm$^{-1}$) but resolvable band. Note that the alkyne Raman spectra of both isomers appear a small "splitting" ($\sim4$ cm$^{-1}$) in 1064 nm pumped Raman spectroscopy, which might be related to excitation wavelength-sensitive Raman responses of the parallel and anti-parallel conformers[1]. The detailed underlying mechanism of this phenomena awaits further investigation. Nonetheless, the following SRS spectra do not show resolvable "splitting" of the alkyne vibrations. The conversion efficiency of the crystalline samples under spontaneous Raman setting is estimated to be $\sim10\%$ under UV irradiation (Supplementary note 2), which might be due to the two-photon excitation of continuous 1064 nm Raman pump during data acquisition that shifts the equilibrium to partially completed photo-cyclization (Supplementary Fig. 4). In contrast, the reverse process of photo-cycloreversion (toward the open-ring structure) reaction under visible irradiation appears much more complete, with no residual Raman signatures of the closed-ring isomers.

### Switchable SRS spectroscopy and microscopy.
Stimulated Raman scattering properties of the engineered probes were measured with SRS microscopy as described in "Methods" and our previous works (Supplementary Fig. 5)[25,26]. SRS spectra were taken in the hyperspectral imaging mode with much shorter pixel dwell time (2 μs) and lower excitation power (15 mW). As expected from spontaneous Raman results, UV irradiation induces the same amounts of frequency redshift of SRS spectra (Fig. 1b and c). Interestingly, SRS appears to be more suitable for photo-switching measurements of these photoactive probes. First of all, no obvious peak splitting could be resolved in either isomers, which may be partially due to the lower spectral resolution of SRS ($\sim12$ cm$^{-1}$) and partly due to the difference in pump wavelength (845 nm). Second, UV conversion efficiency tends to stay higher in SRS, because the backward photo-cycloreversion is less efficient under very short pixel irradiation time (2 μs) in fast laser scanning/imaging mode. In addition, SRS intensity of the new peak (closed-ring isomer) benefits more amplification from the electronic pre-resonance effect[16], because the absorption spectrum of the closed-ring isomer has redshifted (centered $\sim600$ nm) closer toward the SRS pump wavelength (Supplementary Fig. 1b). Consequently, a strong redshifted and quasi-single peak SRS lineshape could be reversibly generated/eliminated by UV/visible irradiation. Furthermore, the sharp SRS spectra of both isomers of DTE-Ph and DTE-TMS are distinguishable, enabling multiplexed imaging with high chemical specificity.

Harnessing the above SRS spectral properties, photo-switchable SRS imaging could be realized by taking images at the newly generated Raman frequencies (Fig. 1b and c). For example, we

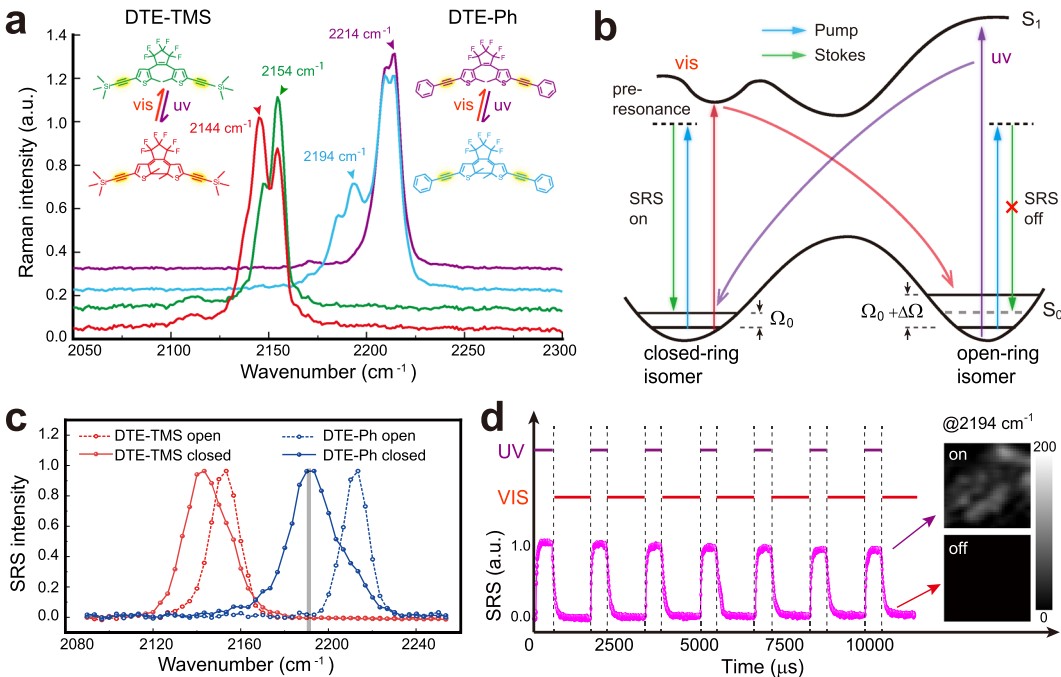

**Fig. 1 Spontaneous and stimulated Raman response of the photoisomerization process of the DTE-alkyne derivatives. a** UV (360 nm) induced photo-cyclization converts the open-ring isomers (green and purple) to the closed-ring structures (red and blue), resulting in spectral red-shifts of the alkyne spontaneous Raman peak (FWHM ~12 cm$^{-1}$). Visible (633 nm) irradiation yields the reversed process of photo-cycloreversion with open-ring isomers and recovered Raman spectra (FWHM ~ 7 cm$^{-1}$). Note that partial photo-cyclization under UV irradiation may results from the two-photon effect of the 1064 nm Raman pump. **b** Schematic diagram of the photoisomerization associated SRS switching with fixed detection frequency ($\Omega_0$) of the closed-ring isomer. **c** SRS spectra of DTE-TMS and DTE-Ph in the open- and closed-ring forms. **d** SRS signal of DTE-Ph acquired at the UV-induced Raman frequency (2194 cm$^{-1}$) shows on/off switching behavior under UV/visible pulsed irradiations.

define the open-ring DTE-Ph as the "off" state and the closed-ring isomer as the "on" state. SRS imaging of DTE-Ph at 2194 cm$^{-1}$ shows clear reversible on/off behavior when irradiated with periodic and out-of-phase UV (360 nm) and visible (633 nm) pulses (Fig. 1d and Supplementary Movie 1), with on-off ratio of ~50. The molecules also present long-term stability and fatigue resistance after numerous photo-switching events (Supplementary Fig. 6). Note that continuous SRS imaging of the "on" state molecules results in gradual "bleaching" of the SRS signal by slow backward conversion (Supplementary Fig. 7), even though SRS uses near-infrared (NIR) wavelengths. Fortunately, this bleaching process is slow enough to cause negligible reduction of SRS intensities under rapid imaging mode and the backward conversion could be largely suppressed by joint UV irradiation during SRS measurements of the closed-ring isomer (Supplementary Fig. 7).

**Photo-rewritable patterning**. Rewritable patterning with SRS microscopy could thus be accomplished by scanning UV focal spot in specific areas with programmed patterns, as well as erasing them with visible irradiation. On uniformly coated DTE-Ph/PMMA thin film, simple square patterns with different sizes and UV irradiation times could be printed and imaged with SRS at 2194 cm$^{-1}$. Note that not only the layered patterns could be readily seen, the layer-dependent SRS intensity and spectra agree with the increased conversion efficiency with longer UV irradiation time (Fig. 2a and b). A rough estimate revealed a pre-resonance amplification ratio of ~8 fold between the "on-state" (closed-ring) at 2194 cm$^{-1}$ and the "off-state" (open-ring) at 2214 cm$^{-1}$ (Supplementary note 2, Supplementary Fig. 8 and Supplementary Table 1). Hence, a conversion efficiency of ~21% has resulted in almost twice the SRS intensity as shown in layer 4 (Fig. 2a and b, Supplementary Table 1). Note that with

continuous UV irradiation, the measured SRS spectrum of DTE-Ph (Fig. 1c) gives an estimate of ~38% closed-ring isomer under the equilibrium between UV photo-cyclization and two-photon excited photo-cycloreversion (Supplementary Fig. 9). To further demonstrate spectral multiplexing with photo-switchable SRS microscopy, we designed a PMMA thin film with spatially separated DTE-Ph and DTE-TMS (Fig. 2c), on which a letter "M" was sketched by UV scan across the border of the two areas. The molecular differences within the printed letter could be resolved by dual-color SRS imaged at 2194 cm$^{-1}$ and 2140 cm$^{-1}$ (Fig. 2d), and hyperspectral SRS revealed clear spectral shifts along the line-cuts across the two parts of the letter (Fig. 2e and f). Although the spatial resolution of the combined UV writing and NIR reading was not optimized (largely due to chromatic aberration), these results demonstrate potential applications of the switchable Raman probes in data memory and information encryption with vibrational multiplexing.

**Painting and erasing in living cells**. Biocompatible DTE-alkyne probes were produced by attaching a mitochondria targeting functional group (pyridinium salt) to the end of the Raman probe (Supplementary Figs. 1a, 10 and Supplementary note 1). We named functionalized DTE-Ph molecules as DTE-Ph-Mito and verified its mitochondrial targeting specificity using a standard mitochondrial dye (Supplementary Fig. 11). High resolution multi-channel SRS images of the "off-state" DTE-Ph-Mito labeled cells could be acquired at different Raman frequencies: on-resonant 2214 cm$^{-1}$ appears bright, off-resonant 2194 cm$^{-1}$ appears dark, 2850 cm$^{-1}$ for lipids and 2930 cm$^{-1}$ for proteins (Fig. 3a), demonstrating chemical specificity of SRS and efficient labeling of DTE-Ph-Mito. Note that a few intracellular "dots" remains in the off-resonant image (Fig. 3a, 2194 cm$^{-1}$), which are most likely from the transient absorption signal of pigments or

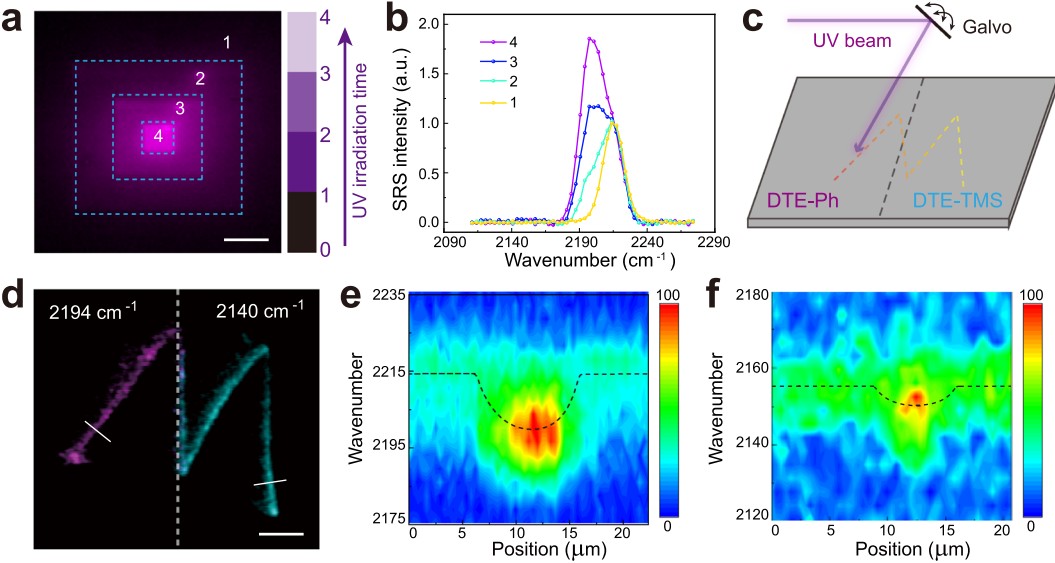

**Fig. 2 Photo-switchable SRS patterning. a** UV printed square areas with increasing irradiation time (0-150 μs/pixel) exhibit increased SRS intensity (at 2194 cm$^{-1}$) and shifted spectra as shown in (**b**). **c** UV beam written letter "M" across the border of DTE-Ph and DTE-TMS coated areas allows dual-color SRS imaging of the two sub-portions of the letter (**d**) with varying SRS spectra along the line-cuts in the DTE-Ph (**e**) and DTE-TMS (**f**) regions. Scale bars: 10 μm in (**a**) and 20 μm in (**d**).

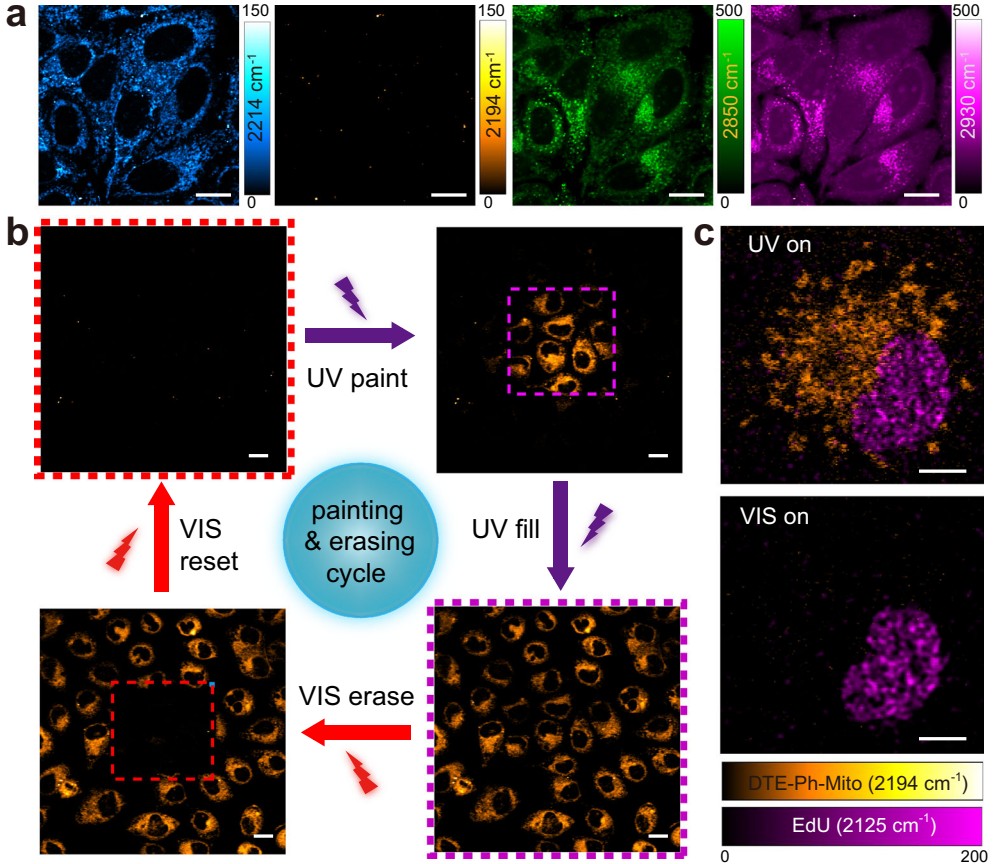

**Fig. 3 Live cell painting and erasing with SRS microscopy. a** DTE-Ph-Mito labeled live HeLa cells imaged at different Raman frequencies: alkyne stretches of the open- (2214 cm$^{-1}$) and closed-ring (2194 cm$^{-1}$) isomers, CH$_2$ symmetric stretch (2850 cm$^{-1}$) and CH$_3$ symmetric stretch (2930 cm$^{-1}$). **b** The cell painting and erasing cycle, through the processes of painting a subset of the cells, lighting (filling) up the whole FOV, erasing a subset of the cells and resetting the whole FOV to the "off" state, using controlled UV/visible irradiations in the purple/red dashed squares. **c** Dual-color SRS imaging of DTE-Ph-Mito (mitochondria targeting) and EdU (nucleus targeting) co-cultured cells under different irradiation conditions. Scale bars: 5 μm in (**c**), 10 μm in (**a**), and 20 μm in (**b**).

metabolites of the SRS dyes. In addition, DTE-Ph-Mito exhibits strong fluorescence (Supplementary Fig. 12), and its two-photon excited fluorescence (TPEF) could be simultaneously imaged, and found to co-localize with corresponding SRS images (Supplementary Fig. 13).

To realize the photo-switching capability in living cells labeled with DTE-Ph-Mito, we set SRS detection frequency to 2194 cm$^{-1}$ and cycle through repeated UV-visible irradiation periods. At the beginning of each cycle, all the cells within the whole field of view (FOV) were reset to the "off" state by visible irradiation. Then a subset of the FOV was turned on by UV scanning of the zoomed-in area (Fig. 3b, purple dashed square). Next, the full FOV was switched on by UV irradiation, uniformly lighting up all the labeled cells. Subsequently, a subset of the cells was switched off by visible light scanning in the zoomed-in area (Fig. 3b, red dashed square), burning a "hole" in the FOV. Finally, the full FOV was erased and returned to the "off" state with visible irradiation, entering the next photo-conversion cycle. The cycling results verified that these molecules could serve as effective photo-switchable SRS probes for live cell imaging, and may hold potentials for cell painting and sorting[27].

Furthermore, we demonstrated a control experiment with dual-color SRS imaging of live cells co-cultured with DTE-Ph-Mito and 5-ethynyl-2′-deoxyuridine (EdU). EdU is a known alkyne-tagged nucleotide labeling newly synthesized DNA, convenient for Raman and SRS studies[28-30]. As expected, the DTE-Ph-Mito labeled mitochondria showed photo-switchable behavior at 2194 cm$^{-1}$. In strong contrast, the nucleus with enriched EdU (at 2125 cm$^{-1}$) remained unchanged, resistant to UV/visible irradiation (Fig. 3c). It could thus be implied that the photoactive nature of these probes expands the multiplexing capability of SRS by spectral modulation with extra light, especially when interfered with large background or overlapping spectra from other inert Raman bands.

**Intracellular pulse-chase experiment**. Photoactive probes are useful for tracking the dynamic motions of intracellular proteins/organelles by pulse-chase measurements, usually with fluorescence-based microscopies[31,32]. Here, we demonstrated similar capability of SRS in DTE-Ph-Mito labeled HeLa cells imaged at 2194 and 2214 cm$^{-1}$ for mitochondria coupled with the "on" and "off" state probes, respectively. The overall distribution of mitochondria could be visualized by resetting the cells to the "off" state with visible irradiation and imaged at 2214 cm$^{-1}$, preparing for the pulse-chase process (Fig. 4a). Next, a confined subset of mitochondria was switched on by UV provoking in a small area, pulsing the initial distribution of mitochondria tagged with the "on" state probes (Fig. 4b, red arrows). The cells were kept in dark and normal culturing conditions under the microscope for certain amount of time, and subsequently imaged with SRS to chase the new distributions of these tagged mitochondria. It could be seen that within 15 min, the originally localized mitochondria in living cells have diffused to a much larger area in the cytoplasm through fission, fusion and dynamic motion (Fig. 4c, red arrows), agreeing with previous observations[33,34]. In strong contrast, in fixed cells, the "on" state labeled mitochondria remain confined in the initial area without observable diffusion. It is also worth mentioning that mitochondria image appearance (resolution and morphology) is related to the labeling density/time of the probe molecules as shown in Supplementary Fig. 14 and previous study[35]. While low labeling density results in well-resolved mitochondria (Fig. 3a and c), they appeared more diffused under relatively high labeling density to achieve sufficient SRS signal for the pulse-chase experiments (Fig. 4).

## Discussion

SRS microscopy was initially developed as a label-free chemical imaging technique without the need of exogenous labeling

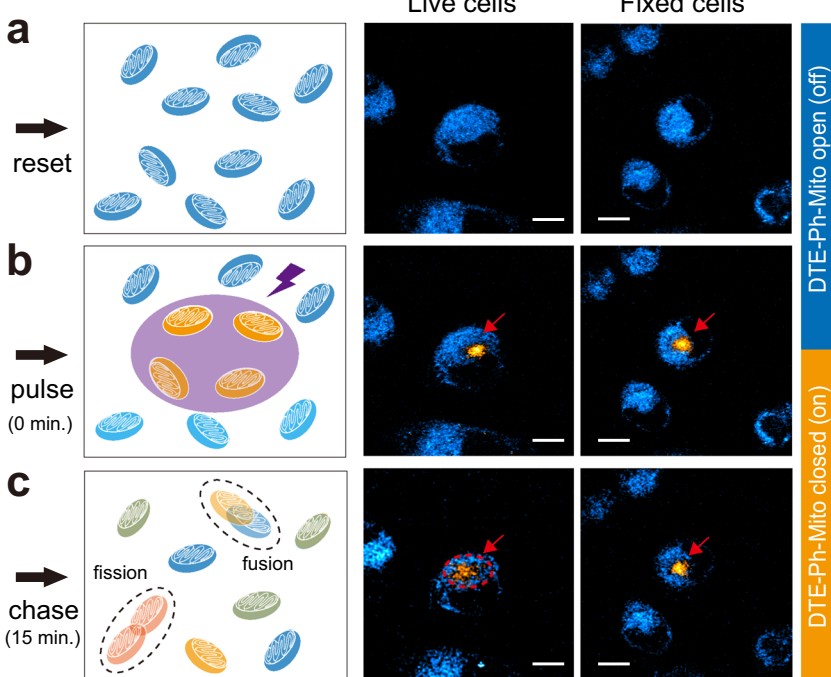

**Fig. 4 Intracellular pulse-chase experiments of mitochondria diffusion with SRS. a** DTE-Ph-Mito labeled cells were prepared in the initial "off" state. **b** A small confined portion of mitochondria were selectively turned on (pulsed) by UV irradiation, and imaged with dual-color SRS to map out both the "off" (cyan) and "on" (yellow) state molecules. **c** After 15 min, the changes of the "on" state mitochondria were imaged (chased). Fixed cells were used as the control group. Scale bar: 10 μm.

molecules. Such "label-free" advantage has allowed unique applications in microspectral analysis[26], 3D chemical profiling[25], and rapid tissue histology for disease diagnosis by mapping the distributions of intrinsic biomolecules, such as lipids, proteins and DNA[12,13,36]. However, insufficient sensitivity and chemical specificity have hindered conventional SRS microscopy for studies in molecular and cell biology. More recent advances in Raman tags have extended SRS to probe labeled small biomolecules for metabolic studies[37–39], and break "multiplex ceiling" for super-multiplexed imaging[16,17]. Molecular engineering by conjugating local vibrational reporters with dyes and fluorescence molecules has further bridged the worlds of fluorescence and Raman scattering, exploiting additional physical properties including electronic pre-resonance Raman/SRS and stimulated Raman excited fluorescence for enhanced vibrational sensitivity[16,18]. Our work has further gained SRS with photoswitching capability, by taking advantage of the existing photochromic molecule (DTE) that alters the spectra of conjugated vibrational reporters. Alkyne derivatives are chosen as Raman reporters because of the sharp single-peak lineshape in the cell silent window, large Raman scattering cross-section, and sensitive to conjugation strength to generate sufficient spectral shifts. Therefore, narrow-band SRS microscopy focused on the newly generated Raman peak could be effectively switched on/off by the same photons that trigger the isomerization of DTE.

Ideal photo-switching probes are expected to the have following properties: (1) good thermal stability for both isomers; (2) high fatigue resistance for repeated photo-switching events; (3) fast switching speed. The DTE-Ph and DTE-TMS synthesized in our study are thermally stable in both open- and closed-ring isomers, just like most DTE derivative reported in the literature[1,24]. Long-term switching capability was demonstrated (Supplementary Fig. 6), showing high fatigue resistance (>80% after 100 continuously switching cycles). At the molecular level, both the photo-cyclization reaction for ring closing and photo-cycloreversion reaction for ring opening processes were found to occur within several picoseconds as measured by time-resolved ultrafast spectroscopy and theoretical calculations[40]. Our measured transition kinetics of SRS intensity triggered by UV pulses revealed sub-millisecond switching time (Supplementary Fig. 15), comparable to most photo-switching fluorescent molecules[41,42].

A major drawback of the current Raman probes is the bleaching caused by SRS beams (845 nm for pump and 1040 nm for Stokes), most likely through two-photon excited cycloreversion to the open-ring isomer ("off" state) as shown in Supplementary Fig. 9. Although very slow, this bleaching process is detrimental to continuous and prolonged SRS imaging, such as the pulse-chase experiments for long-term observations of organelle motions. We also noticed that fusing with biocompatible functional group, DTE-Ph-Mito tends to have increased SRS bleaching rate. Therefore, it is worth future efforts to optimize this family of probes to fully exploit the advantages of vibrational spectroscopy in photo-switchable imaging. Firstly, SRS sensitivity shall be optimized with tailored alkyne derivatives and functional groups[43], which could help reduce the labeling density, SRS laser power, as well as the two-photon bleaching rate. Second, a series of switchable SRS probes with resolvable Raman peaks need to be achieved to enable super-multiplexing, and ideally super-resolution SRS with the principle of RESOLFT microscopy. Furthermore, with our design principle, other types of photochromic molecules and local chemical bonds may also be found for vibrational switching, which not only apply to Raman based microscopy modalities, but are also expected to benefit IR based imaging techniques[44,45].

In conclusion, we have demonstrated photo-switchable SRS microscopy with alkyne-tagged photochromic probes for information

patterning and live cell imaging. Our work holds promise for super-resolution SRS imaging, 3D optical memory/switch and frequency multiplexed storage with vibrational contrasts.

## Methods

**Chemical synthesis**. Methods for chemical synthesis and characterization of new compounds can be found in the Supplementary Note 1.

**Microscope setup**. Our imaging microscope was based on a conventional SRS microscope coupled with additional visible and UV laser source (Supplementary Fig. 5). For the SRS microscope, pulsed femtosecond laser beams from a commercial OPO (optical parametric oscillator) laser (Insight DS+, Newport, CA) were used as the laser source. The fixed fundamental 1040 nm laser was used as the Stokes beam, while the tunable OPO output (680-1300 nm) was served as the pump beam. Chirped by SF57 glass rods, pulse durations of the pump and Stokes beams were stretched to ~2.3 ps and ~1.2 ps, respectively. The intensity of the 1040 nm Stokes beam was modulated at 1/4 of the laser pulse repetition rate ($f_0 = 80$ MHz) using an electro-optical modulator (EOM). The two pulse trains were spatially and temporally overlapped through dichroic mirror and delay stage, delivered into a laser scanning microscope (FV1200, Olympus), and focused onto the sample with an objective lens (Olympus, UPLSAPO 60×W, NA = 1.2). The forward-going pump and Stokes beams after passing through the samples were collected in transmission with a high-NA condenser lens (oil immersion, 1.4 NA, Nikon). The stimulated Raman loss (SRL) signal was optically filtered (CARS ET890/220, Chroma), detected by a homemade reverse-biased photodiode (PD) and demodulated with a lock-in amplifier (LIA) (HF2LI, Zurich Instruments) to feed the analog input of the microscope to form images. In order to induce the conversion of the DTE molecules, additional HeNe laser was served as visible beam, while a solid state diode laser (UV-FN-360, Changchun New Industry Optoelectronic Technology Co., China) provided the necessary UV beam. Visible and UV beams were coupled into the path of pump and Stokes beams through dichroic mirrors. All laser powers were measured after the objective lens. For experiments in Figs. 1 c–d and 2, the Stokes laser power was 10 mW and the pump laser power was 5 mW, while the VIS and UV power were kept at 2–50 μW. For cell imaging (Figs. 3 and 4), the laser powers were used as $P_{Stokes} = 40$ mW and $P_{pump} = 20$ mW, $P_{VIS} = 20$ μW and $P_{UV} = 10$ μW. Pixel dwell times were set at 2 μs for Figs. 2a, 2d, and 4, 10 μs for Fig. 3a and 20 μs for Fig. 3b–c. Image and data analysis were done with commercial software ImageJ and Origin.

**Spectroscopy measurement**. Spontaneous Raman spectra were collected with a confocal Raman microscope (LabRAM HR Evolution, HORIBA) in dark room to avoid ambient light. The DTE powder samples were excited through a 50× air NIR objective (HC PL Fluotar, 0.55 NA, Leica) by a 1064 nm laser (75 mW after the objective). The acquisition time was 10 s for each spectrum. UV-Vis spectra and photoluminescence spectra were measured on Shimadzu UV-2550 spectrophotometer and Hitachi F-4500 fluorescence spectrometer with DMSO solution ($1×10^{-4}$ M), respectively.

**Fabrication of DTE films**. DTE powder (~1 mg) was dissolved in 0.5 ml Poly (methyl methacrylate) (PMMA) and the mixture was spin coated onto quartz substrate and baked at 170 °C for 10 min. To achieve frequency encryption (Fig. 2c–d), the substrate was divided into two parts. Each side was spin coated with corresponding mixture (left: DTE-Ph; right: DTE-TMS) while keeping opposite cover with tape.

**Cell culturing and imaging**. Before labeling experiments, HeLa cells were cultured with DMEM medium (Invitrogen, 11965092) supplemented with 10% FBS (Invitrogen, 16,000) and 1% penicillin–streptomycin (Invitrogen, 15140) at a humidified environment at 37 °C and 5% $CO_2$. All samples were assembled into a chamber using homemade punched slide filled with PBS solution for imaging.

**Colocalization imaging in HeLa with DTE-Ph-Mito and Mito-tracker**. Cells were seeded and cultured on a glass coverslip in a 12-well plate for 24 h and co-stained with Mito-Tracker Green (50 nM) and DTE-Ph-Mito (4 μM) in culture medium for 20 mins. Before imaging, cells were washed with PBS three times (Supplementary Fig. 11).

**SRS imaging in live HeLa cells with DTE-Ph-Mito**. Cells were seeded and cultured on a glass coverslip in a 12-well plate for 24 h and then incubated with 5 μM DTE-Ph-Mito for 1 h. Before imaging, cells were washed with PBS three times (Fig. 3a and b; Supplementary Fig. 13).

**Dual-color imaging in live HeLa cells with DTE-Ph-Mito and EdU**. Cells were first seeded on coverslips with DMEM culture medium for 24 h. DMEM culture medium was then changed to pure DMEM medium without FBS. One day later, medium was replaced back to DMEM culture medium with 100 μM EdU for 22 h. After EdU labeling, cells were wash with PBS three times and further incubated

with 5 μM DTE-Ph-Mito for 1 h. Before imaging, cells were washed with PBS three times (Fig. 3c).

**Pulse-chase experiment with DTE-Ph-Mito.** Cells were seeded and cultured on a glass coverslip in a 6-well plate for 24 h and then incubated with 20 μM DTE-Ph-Mito for 2 h. For control group, cells were then fixed with 4% paraformaldehyde (PFA). Before imaging, cells were washed with PBS three times (Fig. 4).

**Statistics and reproducibility.** Error bars in Supplementary Fig. 10 indicate SD. For all experiments presented as representative images, biological replicates were performed as following: Fig. 3a–c, ten biological replicates each; Fig. 4a–c, two biological replicates each; Supplementary Fig. 11a–c, five biological replicates each; Supplementary Fig. 13, fifteen biological replicates; Supplementary Fig. 14a–c, one biological replicate each.

**Reporting summary.** Further information on research design is available in the Nature Research Reporting Summary linked to this article.

## Data availability

All the data supporting the findings of this study are provided in Supplementary Note 1–2, Supplementary Figures 1–15 and Supplementary Table 1, and are available from the corresponding author upon request.

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

## Acknowledgements

M.J. acknowledges financial support from the National Nature Science Foundation of China (61975033), Shanghai Municipal Science and Technology Major Project (2017SHZDZX01, 2018SHZDZX01) and ZJLab, and Specialized Research Project of the Shanghai Health and Family Planning Commission on Smart Medicine (2018ZHYL0204). C.W. acknowledges financial support from Shenzhen Science and Technology Innovation Commission (KQTD20170810111314625), the National Natural Science Foundation of China (81771930), and the National Key Research and Development Program of China (2018YFB0407200). S.P. acknowledges financial support from the National Research Foundation of Korea (2019-R1A6A1A11044070).

## Author contributions

J.A. performed the SRS microscopy, spontaneous Raman spectroscopy, biological studies and analyzed the data with the help of X.M. and J.L.; X.F. performed the chemical synthesis and the UV-Vis, photoluminescence spectroscopy. H.K. performed the DFT simulation together with S.P.; M.J. and C.W. conceived the concept. J.A., X.F., C.W. and M.J. designed the experiments and wrote the manuscript with the inputs from all authors.

## Competing interests

The authors declare no competing interests.
