## [Peer Review File · Nature Communications]

REVIEWER COMMENTS

Reviewer #1 (Remarks to the Author):

The manuscript reports photo-switchable Raman imaging based on vibrational photochromism of diarylethene with alkyne Raman tags. With stimulated Raman scattering (SRS) microscopy, the authors demonstrate switching on and off the Raman signal upon UV and visible light irradiation. The idea of this work comes from photo-switchable fluorescent dyes, which are essential for fluorescence super-resolution imaging. They claim that the "multiplex ceiling" in conventional fluorescence microscopy due to spectral broadness of fluorescence can potentially be overcome in their Raman-based photo-switchable imaging. They report proof-of-concept photo-switchable imaging of polymer films and cells (both live and fixed) stained with the dye. Also, as a biologically meaningful application, they performed pulse-chase imaging, in which time-dependent change of mitochondrial distribution is visualized by monitoring how photo-switched dyes are attached to mitochondria changes after local photoswitching. Overall, the manuscript is well written and organized to support their conclusion. It is a nice piece of work. For this reason, I recommend publication of the manuscript in *Nature Communications*. However, the authors should address the following issues.

1. It is not explicitly explained in the manuscript whether Figure 1a shows spontaneous Raman or SRS spectra. This should be clarified.
2. In Figure 1b, it looks like no SRS signal is generated due to the absence of electronic resonance enhancement. However, the non-negligible Raman signal is detected from the open-ring isomer as shown in Fig. 1a. Presentation in Fig. 1b needs to be changed.
3. In Figure 1d, UV irradiation stops before the SRS intensity saturates. I wonder if a higher SRS signal is obtained by longer UV irradiation or side-effect such as bleaching occurs if the dye is exposed to UV for longer time.
4. In Figure 1, it would be useful to clarify in the caption that the spontaneous Raman spectra shown for the UV form are showing a partial photo-cyclization to the UV form and not a pure open-form spectrum.
5. In Figure 2e and 2f, the width of the switched area looks broader than the diffraction limit. Discussion about factors that determine the size of switching area should be provided.
6. On Page 8, the DTE-alkyne probes are addressed as "Biocompatible", but no viability tests are done on the cells in the presence of the compound to make this claim.
7. In Figure 3a, the text describes it as "on-resonant 2214 cm^{-1} appears bright, off-resonant 2194 cm^{-1} appears dark" though there remain visible bright dots in the "off" image, which look like lipid droplets in the periphery of the cell. These small bright dots are also visible in all the off states of Figure 3. Interestingly, Figure S9 also shows strong DTE-Ph-Mito SRS signal in what seems to be lipid droplets that do not show TPEF signal. The reason why those dots are detected only in SRS images, not in fluorescence images, should be discussed. The reference reporting solvatochromism with an alkyne labelled Raman dye with some similarities (Gala de Pablo, J.; Chisholm, D. R.; Steffen, A.; Nelson, A. K.; Mahler, C.; Marder, T. B.; Peyman, S. A.; Girkin, J. M.; Ambler, C. A.; Whiting, A.; Evans, S. D. *Analyst* 2018, 143 (24), 6113–6120.) could be useful to understand this effect.
8. The paper assumes that the DTE-Ph-Mito dyes target mitochondria without direct evidence. The authors should compare their SRS-based mitochondria images with those with a standard fluorescence dye for mitochondria imaging, such as mitotracker, to confirm their newly synthesized dye targets mitochondria.
9. The absorption spectra from Figure S1 only shows absorbance until 800 nm. If there is in fact no absorbance past this point, the partial photo-cyclization seen when using the 1064 nm laser beam for spontaneous Raman (Figure 1, discussed in page 5) must have likely been due to the high power used (75 mW), causing conversion due to two-photon absorption. Doing a longer acquisition time at a lower power in the presence of UV light (5-10 mW) may solve this issue and show the high resolution

spectrum of the compound UV form without the signal of the visible form (as it follows from the SRS results on figure S7).

10. On page 6, the meaning of "with on-off ratio of 50" is unclear and should be clarified.

11. The time scale should be shown in Figure 1d.

12. The color scale in Figure 2a should be from 0, not 1 as they assume that Layer 1 is dominated by the off conformer.

13. A label is needed for the color scale in Figure 2e and f.

14. In the caption of Supplementary Table 1, "Figure 2d" should be "Figure 2a".

15. In Supplementary Note 2, they discuss layers upside down. For example, layer 1, not 4, is the layer without UV irradiation. Also, they should refer to Fig. 2a-b, not 2c-d in this section.

Typos:

Page 5, line 12: reverse -> reverse

Line 20: mw -> mW

Page 13, line 16: simulated -> stimulated

Supplementary note 2: viogt -> Voigt; close-form -> closed-form; from all layer -> from all layers

Figure S6 caption: interactive -> interact

Page 10: labeled -> labelled

Page 15: for three times -> three times

Reviewer #2 (Remarks to the Author):

This is a highly novel work in the field of coherent Raman scattering microscopy. The development of photoswitchable Raman tags represents a significant advance in functional imaging of living cells. I have a comment on the resolution and/or specificity in figure 3 and figure 4 regarding the DTE-Ph-Mito dye. This probe is expected to label mitochondria. Yet the resolution in figure 3 and figure 4 seems not sufficient. The dye fills the entire cytoplasm, not resembling the distribution of mitochondria inside a cell in figure 3 and a small area in figure 4. I suggest that the authors label mitochondria with a standard dye such as rhodamine123 and compare the SRS contrast to fluorescence contrast inside the same cell.

Reviewer #3 (Remarks to the Author):

This manuscript primarily reports the design and characterization of the basic vibrational photo-switching properties of specifically engineered Raman probes, where alkyne groups as well-established biorthogonal Raman tags have been introduced into photochromic diarylethene. Contrary to their explicitly stated motivation (e.g. in their current version of the Abstract), the authors have not demonstrated yet the high potential of photochromic probes with reversible Raman responses for surmounting the current limitations inherent to single-molecule fluorescence spectroscopy and super-resolution fluorescence microscopies, such as the lack of sufficient chemical specificity of switchable fluorophores and their susceptibility to photo-bleaching. Rather, the authors' novel photochromic vibrational probes have been successfully applied to switchable SRS microscopy, demonstrating rewritable information patterning, controlled activation, erasing and chasing in living cells, so far, on the proof-of-principle level. Even so, I consider the current work as an important contribution to the design and application of photo-switchable vibrational probes.

For most parts, the experimental work was thoroughly performed, and the manuscript was clearly written. I recommend publication in Nature Communications provided the authors will have revised their manuscript in order to address the minor deficiencies and open questions that I have listed below

in more detail. Please understand my comments as to strengthen any revisions of this manuscript:

1) Introduction, line 45: The original seminal literature on the development of SRS microscopy should be credited!

2) Results paragraph starting with line 81: Besides the peak positions, please also provide the measured linewidths of all spontaneous Raman features shown in Figs. 1a and S3.

3) Fig. 1b and its discussion (lines 245-263): In my view, the proposed schematic diagram of the photoisomerization associated SRS switching cannot reflect the full picture! As observed by the authors themselves (!), processes like multi-photon excitations by the pump and Stokes pulses alone (for example, see: TPEF data in Fig. S9, photo-switching of the photoactive Raman peak at 2194 cm^{-1} under "normal" SRS conditions in Fig. S6, and Ref. [34] on "One- and multi-photon cycloreversion reaction dynamics of diarylethene derivative ...") have simply not been taken into account, yet. In this context, a more detailed discussion regarding the conversion rates at the chosen photon flux densities of the pump, Stokes, UV, and VIS beams and regarding the dependence of photoisomerization on the pump and Stokes pulse powers would have been beneficial. For example, can you provide estimation for the relative equilibrium ground-state populations of the closed- and open-ring isomers under the chosen experimental "on" and "off" conditions? Please amend your discussion accordingly.

4) Results, Fig. 1d: Please provide quantitative labels for the SRS intensities in both the y-axis of the time-trace-graph and for the SRS images (i.e., LUT). Likewise, show time labels and units on the time-axis.

5) Results paragraph starting with line 165 and Fig. 3: While this study demonstrates the proof-of-principle of "Painting and erasing in living cells", it does not overcome the limitations inherent to similar fluorescence-based experiments. Strictly speaking, such results can be obtained much easier based on conventional fluorescence-based imaging. Therefore, please amend here a more critical discussion of your results that also takes the future steps into account that are required for actually "breaking the multiplex ceiling" advantage potentially offered by photochromic SRS probes.

6) Results paragraph starting with line 200 and Fig. 3: Likewise, the capability and information content of the demonstrated proof-of-principle "Intracellular pulse-chase experiment" do not go beyond those of similar fluorescence-based experiments (as noted by the authors themselves in line 215). Again, what would be needed for future designs of photochromic SRS probes in order to fully exploit the potential benefits of switchable SRS microscopy?

7) Methods paragraph starting with line 275: The pump and Stokes pulses were stretched to picoseconds. How many picoseconds, approximately? And, how do these effective pulse lengths compare with the time scale of "several picoseconds" stated for the photo-cyclization and photo-cycloreversion reactions in line 247?

8) Supplementary Note 2, last paragraph: Shouldn't the reference to the figure showing the spectral decomposition analysis read "Supplementary Fig. 7" (instead of "Supplementary Fig. 5")?

9) Supplementary Fig. S8: For completeness, please also provide the corresponding absorption spectra for DTE-Ph-Mito after UV and visible irradiation (similar to those for DTE-TMS and DTE-Ph shown in Fig. S1b).

We thank the reviewers for their constructive comments, based on which we have revised our manuscript. All the changes have been made in the revised manuscript with highlighted markups. Below are the point-by-point responses to the comments:

Reviewer #1:

The manuscript reports photo-switchable Raman imaging based on vibrational photochromism of diarylethene with alkyne Raman tags. With stimulated Raman scattering (SRS) microscopy, the authors demonstrate switching on and off the Raman signal upon UV and visible light irradiation. The idea of this work comes from photo-switchable fluorescent dyes, which are essential for fluorescence super-resolution imaging. They claim that the "multiplex ceiling" in conventional fluorescence microscopy due to spectral broadness of fluorescence can potentially be overcome in their Raman-based photo-switchable imaging. They report proof-of-concept photo-switchable imaging of polymer films and cells (both live and fixed) stained with the dye. Also, as a biologically meaningful application, they performed pulse-chase imaging, in which time-dependent change of mitochondrial distribution is visualized by monitoring how photo-switched dyes are attached to mitochondria changes after local photoswitching.

Overall, the manuscript is well written and organized to support their conclusion. It is a nice piece of work. For this reason, I recommend publication of the manuscript in Nature Communications. However, the authors should address the following issues.

Response: We very much appreciate the reviewer for the careful review and positive comments.

1. It is not explicitly explained in the manuscript whether Figure 1a shows spontaneous Raman or SRS spectra. This should be clarified.

Response: We thank the reviewer for pointing it out. We have revised the caption of Figure 1a to clarify that these are spontaneous Raman spectra.

2. In Figure 1b, it looks like no SRS signal is generated due to the absence of electronic resonance enhancement. However, the non-negligible Raman signal is detected from the open-ring isomer as shown in Fig. 1a. Presentation in Fig. 1b needs to be changed.

Response: We are sorry for causing the confusion. Figure 1b is meant to show the off-resonance of SRS for the open-ring isomer when the detection Raman frequency remains in resonance with the closed-ring form (e.g. 2194 cm^{-1}), because the open-ring isomer has blue-shifted Raman peak (2214 cm^{-1}). The spectral shifts/changes are shown in Figure 1a. The narrow-band detection of SRS combined with the large enough spectral shift induced by photo-isomerization have enabled the switching of SRS signal at the particular frequency, i.e. the peak frequency of the closed-ring isomer. The electronic pre-resonance effect happens to enhance the spectral difference and on/off ratio between the two isomers. We have slightly modified Figure 2b and its caption to

clarify it.

3. In Figure 1d, UV irradiation stops before the SRS intensity saturates. I wonder if a higher SRS signal is obtained by longer UV irradiation or side-effect such as bleaching occurs if the dye is exposed to UV for longer time.

Response: We thank the reviewer for bringing up this issue. In the previous results, UV irradiation was set at the near-saturation point, slightly higher SRS signal could be reached at saturation with longer UV irradiation time as shown in the revised Figure 1d, as well as the single cycle data shown below. Prolonged UV irradiation is more likely to cause irreversible reactions to generate un-switchable byproducts as indicated in Ref. 1.

Figure R1. A typical single-cycle transition of SRS signal under UV/visible irradiation.

4. In Figure 1, it would be useful to clarify in the caption that the spontaneous Raman spectra shown for the UV form are showing a partial photo-cyclization to the UV form and not a pure open-form spectrum.

Response: The partial photo-cyclization under UV is mostly due to the two-photon effect of the Raman excitation beam (1064 nm) as shown in the revised Supplementary Figure 4. We have also revised the caption of Figure 1 and the main text to clarify this.

5. In Figure 2e and 2f, the width of the switched area looks broader than the diffraction limit. Discussion about factors that determine the size of switching area should be provided.

Response: We thank the reviewer for the insightful observation. The spatial resolution of the switching based writing/reading is mainly limited by several factors: a) Size of the UV focal spot. Since the microscope objective (Olympus, UPLSAPO 60×W) was only optimized for the near IR range, large chromatic aberration is expected to exist that enlarges the UV spot at the image plane of SRS. b) The single-photon nature of the UV induced photo-isomerization process may broaden the switching area under longer

irradiation time, due to the accumulated effect of scattered UV photons. Although Figure 2e and f in this work are not aiming at optimizing the spatial resolution of UV writing, these issues are critical for our on-going work on switching based super-resolution SRS imaging. Therefore, we are still working on optimizing the spatial mode of UV beam, minimizing the chromatic aberration for both UV and NIR, as well as tweaking the best UV irradiation time. We have added more discussions in the revised manuscript (Line 161-163, Page 8).

6. On Page 8, the DTE-alkyne probes are addressed as “Biocompatible”, but no viability tests are done on the cells in the presence of the compound to make this claim.

Response: We meant to express that the functionalized molecules have better cell permeability compared with the original molecular (DTE-Ph). We have tested the cell viability response to the TE-Ph-Mito as shown below and in Supplementary Fig. 10. Thanks for the advice.

Figure R2. Cell viability tested by MTT assay of DTE-Ph-Mito cultured HeLa cells.

7. In Figure 3a, the text describes it as “on-resonant 2214 cm^{-1} appears bright, off-resonant 2194 cm^{-1} appears dark” though there remain visible bright dots in the “off” image, which look like lipid droplets in the periphery of the cell. These small bright dots are also visible in all the off states of Figure 3. Interestingly, Figure S9 also shows strong DTE-Ph-Mito SRS signal in what seems to be lipid droplets that do not show TPEF signal. The reason why those dots are detected only in SRS images, not in fluorescence images, should be discussed. The reference reporting solvatochromism with an alkyne labelled Raman dye with some similarities (Gala de Pablo, J.; Chisholm, D. R.; Steffen, A.; Nelson, A. K.; Mahler, C.; Marder, T. B.; Peyman, S. A.; Girkin, J. M.; Ambler, C. A.; Whiting, A.; Evans, S. D. *Analyst* 2018, 143 (24), 6113–6120.) could be useful to understand this effect.

Response: These bright dots are mostly likely originated from the transient absorption signal of certain pigments/chromophore in the lysosomes. They might be metabolites

of the SRS probes with light absorption but non-fluorescence. Since they have very broad spectra and do not show vibrational resonance, it is less likely that the underlying mechanism would be related to solvatochromism. We have added discussions in the revised text (Line 175-177, Page 8).

8. The paper assumes that the DTE-Ph-Mito dyes target mitochondria without direct evidence. The authors should compare their SRS-based mitochondria images with those with a standard fluorescence dye for mitochondria imaging, such as mitotracker, to confirm their newly synthesized dye targets mitochondria.

Response: We thank the reviewer for pointing out this important issue. We have performed additional experiments to verify that DTE-Ph-Mito indeed targets mitochondria as shown below. As can be seen, DTE-Ph-Mito stained mitochondria show strong two-photon fluorescence that perfectly co-localized with the Mito-Tracker Green labelled ones, revealing the high specificity of mitochondria targeting of DTE-Ph-Mito. We have added the results in the supplementary materials (Supplementary Fig. 11) and the main text (Line 169-171, Page 8).

Figure R3. Mitochondrial labeling specificity of DTE-Ph-Mito. Two-photon fluorescent images of HeLa cells co-stained with (a) Mito-Tracker Green (50 nM) and (b) DTE-Ph-Mito (4 μ M) in culture medium for 20 min. (c) Merged image of (a) and (b). Excitation wavelength 976 nm for Mito-Tracker Green and 801 nm for DTE-Ph-Mito.

9. The absorption spectra from Figure S1 only shows absorbance until 800 nm. If there is in fact no absorbance past this point, the partial photo-cyclization seen when using the 1064 nm laser beam for spontaneous Raman (Figure 1, discussed in page 5) must have likely been due to the high power used (75 mW), causing conversion due to two-photon absorption. Doing a longer acquisition time at a lower power in the presence of UV light (5-10 mW) may solve this issue and show the high resolution spectrum of the compound UV form without the signal of the visible form (as it follows from the SRS results on figure S7).

Response: We agree that partial photo-cyclization in spontaneous Raman measurement is most likely due to the high Raman pump power used (75 mW), causing conversion due to two-photon absorption. We appreciate the reviewer's suggestion and we did our

best to acquire Raman spectra with more closed-ring isomers. However, the commercial spontaneous Raman spectrometer (Horiba) is unable to couple 1064nm with external UV beam, and the pure closed-ring form could be hardly reached at the current stage. Nonetheless, our SRS results indeed show the trend of more complete photo-cyclization with longer UV irradiation time (Fig. 2a, b and Supplementary Fig. 8).

10. On page 6, the meaning of “with on-off ratio of 50” is unclear and should be clarified.

Response: We have recorded the spectrum of open-ring and closed-ring isomers of DTE-Ph during switching cycles (shown below). Therefore, the SRS intensity of “on” and “off” state at 2194 cm^{-1} could be measured to be ~ 1.776 and ~ 0.035 , respectively. The on-off ratio is then estimated by as $1.776/0.035 \sim 50$.

Figure R4. Typical SRS spectra of open-ring and closed-ring isomers of DTE-Ph.

11. The time scale should be shown in Figure 1d.

Response: We have edited the time scale of Figure 1d in the revised manuscript.

12. The color scale in Figure 2a should be from 0, not 1 as they assume that Layer 1 is dominated by the off conformer.

Response: We are sorry for the confusion. The color scale in Figure 2a was meant to indicate the four layers. We have revised it according to the reviewer’s suggestion.

13. A label is needed for the color scale in Figure 2e and f.

Response: We have added label for color scales in Figure 2e-f in the revised manuscript.

Thanks for the suggestion.

14. In the caption of Supplementary Table 1, "Figure 2d" should be "Figure 2a".

Response: We have modified it in the revised manuscript, thanks for pointing it out.

15. In Supplementary Note 2, they discuss layers upside down. For example, layer 1, not 4, is the layer without UV irradiation. Also, they should refer to Fig. 2a-b, not 2c-d in this section.

Response: We are sorry for the mislabeling and referencing. We have corrected them in the revised Supporting Information.

Typos:

Page 5, line 12: revere -> reverse

Line 20: mw -> mW

Page 13, line 16: simulated -> stimulated

Supplementary note 2: viogt -> Voigt; close-form -> closed-form; from all layer -> from all layers

Figure S6 caption: interactive -> interact

Page 10: labeled -> labelled

Page 15: for three times -> three times

Response: We have corrected all these typos in the revision. Thank you for the careful work!

Reviewer #2:

This is a highly novel work in the field of coherent Raman scattering microscopy. The development of photoswitchable Raman tags represents a significant advance in functional imaging of living cells. I have a comment on the resolution and/or specificity in figure 3 and figure 4 regarding the DTE-Ph-Mito dye. This probe is expected to label mitochondria. Yet the resolution in figure 3 and figure 4 seems not sufficient. The dye fills the entire cytoplasm, not resembling the distribution of mitochondria inside a cell in figure 3 and a small area in figure 4. I suggest that the authors label mitochondria with a standard dye such as rhodamine123 and compare the SRS contrast to fluorescence contrast inside the same cell.

Response: We very much thank the reviewer for the positive comments of our work. We also thank the reviewer for bringing up this important issue, which has encouraged us to perform additional experiments to improve the quality of the work.

- (1) Regarding the labeling specificity of DTE-Ph-Mito, we have conducted fluorescence imaging of mitochondria co-labelled with the standard dye (Mito-Tracker Green) and our SRS dye (DTE-Ph-Mito). Their co-localization was verified and shown in Figure R3. We have added discussion of this in main text

(Line 169-171, Page 8).

- (2) Regarding the resolution issue, it was essentially due to the dyes filling the entire cytoplasm under improper culturing condition. As shown in Figure R5 below, with increased incubation time, the labeling density is enriched in the whole mitochondrial region, lowering the resolving power of individual structure. This phenomenon was also reported in the literature [J Am Chem Soc 139, 17022-17030, (2017)]. We have optimized the labeling condition to achieve better resolved mitochondria images in the revised Figure 3. The pulse-chase experiments in Figure 4 are more challenging, thus it was kept under high incubation concentration (20 μ M) and long incubation time (2h) for the proof-of-principle demonstration. On-going researches on improving the signal intensity and photo-stability of the SRS dyes will greatly help reducing the labeling density and ultimately solve this issue. We have added discussion of this in Lines 220-225, Page 10 and 267-273, Page 13.

Figure R5. Fluorescence images of HeLa cells stained with 10 μ M DTE-Ph-Mito for (A) 30 mins, (B) 60 mins, and (C) 90 mins.

Reviewer #3:

This manuscript primarily reports the design and characterization of the basic vibrational photo-switching properties of specifically engineered Raman probes, where alkyne groups as well-established biorthogonal Raman tags have been introduced into photochromic diarylethene. Contrary to their explicitly stated motivation (e.g. in their current version of the Abstract), the authors have not demonstrated yet the high potential of photochromic probes with reversible Raman responses for surmounting the current limitations inherent to single-molecule fluorescence spectroscopy and super-resolution fluorescence microscopies, such as the lack of sufficient chemical specificity of switchable fluorophores and their susceptibility to photo-bleaching. Rather, the authors' novel photochromic vibrational probes have been successfully applied to switchable SRS microscopy, demonstrating rewritable information patterning, controlled activation, erasing and chasing in living cells, so far, on the proof-of-principle level. Even so, I consider the current work as an important contribution to the design and application of photo-switchable vibrational probes.

Response: We very much appreciate the reviewer's constructive comments, pointing out both the limitations and novelties of our work. Indeed, it is our long-term pursuit to enable switchable SRS for multiplexed photoactive chemical imaging and super-resolution vibrational imaging. We are not there yet, and we are grateful that the reviewer could acknowledge the current progress we have made in demonstrating the proof-of-principle results of photo-switchable SRS probes.

For most parts, the experimental work was thoroughly performed, and the manuscript was clearly written. I recommend publication in Nature Communications provided the authors will have revised their manuscript in order to address the minor deficiencies and open questions that I have listed below in more detail. Please understand my comments as to strengthen any revisions of this manuscript:

1) Introduction, line 45: The original seminal literature on the development of SRS microscopy should be credited!

Response: We are sorry for missing a few original literatures on the development of SRS microscopy. We have added them in the revised manuscript.

2) Results paragraph starting with line 81: Besides the peak positions, please also provide the measured linewidths of all spontaneous Raman features shown in Figs. 1a and S3.

Response: We have added these parameters in revised caption of Figure 1a and Supplementary Fig. 3. Thanks for the suggestion!

3) Fig. 1b and its discussion (lines 245-263): In my view, the proposed schematic diagram of the photoisomerization associated SRS switching cannot reflect the full picture! As observed by the authors themselves (!), processes like multi-photon excitations by the pump and Stokes pulses alone (for example, see: TPEF data in Fig. S9, photo-switching of the photoactive Raman peak at 2194 cm⁻¹ under "normal" SRS conditions in Fig. S6, and Ref. [34] on "One- and multi-photon cycloreversion reaction dynamics of diarylethene derivative ...") have simply not been taken into account, yet. In this context, a more detailed discussion regarding the conversion rates at the chosen photon flux densities of the pump, Stokes, UV, and VIS beams and regarding the dependence of photoisomerization on the pump and Stokes pulse powers would have been beneficial. For example, can you provide estimation for the relative equilibrium ground-state populations of the closed- and open-ring isomers under the chosen experimental "on" and "off" conditions? Please amend your discussion accordingly.

Response: We thank the reviewer for the insightful observation. It is true that multi-photon induced cycloreversion exists during the SRS measurements, as can be seen in the spontaneous Raman spectra of the incomplete photo-cycloreversion under 1064 nm excitation, as well as the gradual decay of SRS signal with increasing imaging time. However, we wish to keep Figure 1b relatively simple and clear, sticking with the main

photoreaction pathways. To provide the more complete picture, we have added a supplementary Fig. 9b to illustrate the multi-photon process. The relative conversion rates between UV, VIS and SRS beams can be readily estimated in the measured transient dynamics shown in supplementary Fig. 7. The two-photon induced cycloreversion by SRS beams ($\tau \sim 800 \mu\text{s}$) is much slower than the VIS photo-cycloreversion ($\tau \sim 70 \mu\text{s}$). Regarding the ground-state population, we can estimate by spectral decomposition of SRS spectra under different conditions. For the “off” state induced by VIS irradiation, all the molecules are in ground state (100%). For the “on” state induced by continuously UV irradiation, the remaining ground state molecule is estimated to be $\sim 38\%$ under SRS imaging. We have added more discussion in Line 152-155, Page 7 to clarify this issue.

4) Results, Fig. 1d: Please provide quantitative labels for the SRS intensities in both the y-axis of the time-trace-graph and for the SRS images (i.e., LUT). Likewise, show time labels and units on the time-axis.

Response: We have edited the labels of Figure 1d in the revised manuscript.

5) Results paragraph starting with line 165 and Fig. 3: While this study demonstrates the proof-of-principle of “Painting and erasing in living cells”, it does not overcome the limitations inherent to similar fluorescence-based experiments. Strictly speaking, such results can be obtained much easier based on conventional fluorescence-based imaging. Therefore, please amend here a more critical discussion of your results that also takes the future steps into account that are required for actually “breaking the multiplex ceiling” advantage potentially offered by photochromic SRS probes.

Response: We thank the reviewer for this critical comment. Indeed, we haven't demonstrated super-multiplexed SRS “painting and erasing” in this work, but rather showed such potential by demonstrating: (1) multi-color photo-switching as shown in Fig. 2d and (2) painting/erasing in living cells. In order to take the full advantage of SRS to “break the multiplex ceiling”, we mainly need to engineer a series of photo-switchable SRS dyes with resolvable Raman peaks, which is in principle achievable by combining the design method in this work and previous alkyne based SRS probes [Nature 544, 465-470 (2017), Nat Methods 15, 194-200 (2018) and Phys Biol 16, 041003 (2019)], such as modifying the alkyne terminal group and isotopic editing. These are on-going researches and we regret not being able to show more advanced results in this work. It is true that the “painting and erasing in living cells” is easier to obtain with fluorescence based methods, but we are not aiming at going beyond fluorescence at the current stage, but to show the first demonstration with vibrational imaging. We have added more discussions of this issue in Line 267-273, Page 13.

6) Results paragraph starting with line 200 and Fig. 3: Likewise, the capability and information content of the demonstrated proof-of-principle “Intracellular pulse-chase experiment” do not go beyond those of similar fluorescence-based experiments (as

noted by the authors themselves in line 215). Again, what would be needed for future designs of photochromic SRS probes in order to fully exploit the potential benefits of switchable SRS microscopy?

Response: Again, we thank the reviewer for the critical comment. In order to go beyond fluorescence based experiments, we need to take these future steps: (1) engineer a series of photo-switchable SRS dyes with resolvable Raman peaks as discussed above; (2) optimize the photostability of these dyes, since the pulse-chase experiments have more strict requirement for stability, especially for long-term dynamical studies; (3) optimize the Raman cross section of the SRS dyes (such as increase the number of alkynyl groups), which would help improve photostability by reducing photon flux in SRS measurements; (4) demonstrate RESOLFT-like super-resolution SRS with super-multiplexing. Although these steps are very challenging, it is our long-term pursuit to fully exploit the advantages of SRS in switchable vibrational imaging. We have added more discussions in Line 267-273, Page 13.

7) Methods paragraph starting with line 275: The pump and Stokes pulses were stretched to picoseconds. How many picoseconds, approximately? And, how do these effective pulse lengths compare with the time scale of “several picoseconds” stated for the photo-cyclization and photo-cycloreversion reactions in line 247?

Response: Femtosecond pulses were stretched to ~1.2 ps for the Stokes and ~2.3 ps for the pump, we have included these numbers in the revised manuscript (Line 293-294, Page 14). According the literature, the photo-cyclization and photo-cycloreversion reactions normally take within 10 ps. However, these time scales are less relevant in our study, because the photo-isomerization reactions were induced by continuous wave (CW) UV and visible light, and the picosecond SRS pulses are only used to probe the signal of the quasi steady-state molecules. These times scales will be relevant if we want to measure the transient response of the molecule by UV/Visible pulse excitation, such as the experiments using femtosecond time-resolved SRS spectroscopy [Annu Rev Phys Chem 58, 461-488 (2007)].

8) Supplementary Note 2, last paragraph: Shouldn't the reference to the figure showing the spectral decomposition analysis read “Supplementary Fig. 7” (instead of “Supplementary Fig. 5”)?

Response: We thank the reviewer for pointing out this mistake. We have corrected it the revised Supplementary Information.

9) Supplementary Fig. S8: For completeness, please also provide the corresponding absorption spectra for DTE-Ph-Mito after UV and visible irradiation (similar to those for DTE-TMS and DTE-Ph shown in Fig. S1b).

Response: The absorption spectra of DTE-Ph-Mito after UV and visible irradiation

have been measured and shown below and revised Supplementary Figure 12b.

Figure R6. The fluorescence (Excitation wavelength: 362 nm) and absorption spectra of DTE-Ph-Mito after UV and visible irradiations.

REVIEWERS' COMMENTS

Reviewer #1 (Remarks to the Author):

The authors have addressed my comments reasonably well. I recommend publication of this work in Nature Communications. It seems that the other two reviewers are also positive about their work. Nice work!

Reviewer #2 (Remarks to the Author):

The authors have addressed my comments nicely. I recommend publication of this manuscript.